# The Impact of miR-155-5p on Myotube Differentiation: Elucidating Molecular Targets in Skeletal Muscle Disorders

**DOI:** 10.3390/ijms25031777

**Published:** 2024-02-01

**Authors:** Letícia Oliveira Lopes, Sarah Santiloni Cury, Diogo de Moraes, Jakeline Santos Oliveira, Grasieli de Oliveira, Otavio Cabral-Marques, Geysson Javier Fernandez, Mario Hiroyuki Hirata, Da-Zhi Wang, Maeli Dal-Pai-Silva, Robson Francisco Carvalho, Paula Paccielli Freire

**Affiliations:** 1Department of Structural and Functional Biology, Institute of Biosciences, São Paulo State University (UNESP), Botucatu 18618-689, Brazil; leticia.oliveira@unesp.br (L.O.L.); santiloni.cury@unesp.br (S.S.C.); dioxiide2@gmail.com (D.d.M.); jakeline.oliveira@unesp.br (J.S.O.); oliveira.grase@gmail.com (G.d.O.); papacriolla@gmail.com (G.J.F.); maeli.dal-pai@unesp.br (M.D.-P.-S.); 2Department of Clinical and Toxicological Analyses, School of Pharmaceutical Sciences, University of São Paulo, São Paulo 05508-000, Brazil; otavio.cmarques@gmail.com (O.C.-M.); mhhirata@usp.br (M.H.H.); 3Department of Immunology, Institute of Biomedical Sciences, University of São Paulo, São Paulo 05508-000, Brazil; 4Network of Immunity in Infection, Malignancy, and Autoimmunity (NIIMA), Universal Scientific Education and Research Network (USERN), São Paulo 05508-000, Brazil; 5Department of Medicine, Division of Molecular Medicine, University of São Paulo School of Medicine, São Paulo 05403-010, Brazil; 6Laboratory of Medical Investigation 29, University of São Paulo School of Medicine, São Paulo 05403-010, Brazil; 7Interunit Postgraduate Program on Bioinformatics, Institute of Mathematics and Statistics (IME), University of São Paulo, São Paulo 05508-090, Brazil; 8College of Medicine, University of Antioquia, UdeA, Medellín 53-108, Colombia; 9Health Heart Institute, Center for Regenerative Medicine, University of South Florida, Tampa, FL 33612, USA; dazhiw@usf.edu

**Keywords:** miR-155, microRNA, non-coding RNAs, muscular dystrophies, DMD, RNA sequencing

## Abstract

MicroRNAs are small regulatory molecules that control gene expression. An emerging property of muscle miRNAs is the cooperative regulation of transcriptional and epitranscriptional events controlling muscle phenotype. miR-155 has been related to muscular dystrophy and muscle cell atrophy. However, the function of miR-155 and its molecular targets in muscular dystrophies remain poorly understood. Through in silico and in vitro approaches, we identify distinct transcriptional profiles induced by miR-155-5p in muscle cells. The treated myotubes changed the expression of 359 genes (166 upregulated and 193 downregulated). We reanalyzed muscle transcriptomic data from dystrophin-deficient patients and detected overlap with gene expression patterns in miR-155-treated myotubes. Our analysis indicated that miR-155 regulates a set of transcripts, including *Aldh1l*, *Nek2*, *Bub1b*, *Ramp3*, *Slc16a4*, *Plce1*, *Dync1i1*, and *Nr1h3*. Enrichment analysis demonstrates 20 targets involved in metabolism, cell cycle regulation, muscle cell maintenance, and the immune system. Moreover, digital cytometry confirmed a significant increase in M2 macrophages, indicating miR-155’s effects on immune response in dystrophic muscles. We highlight a critical miR-155 associated with disease-related pathways in skeletal muscle disorders.

## 1. Introduction

The skeletal muscle is the largest protein reservoir in the body and exhibits high plasticity in response to processes regulating growth, regeneration, metabolism, and atrophy [1,2]. Muscle atrophy is characterized by decreased protein content, muscle fiber diameter, force production, and increased fatigue [3,4,5]. Ubiquitin–proteasome and autophagy–lysosome are the main cellular degradation systems that regulate muscle atrophy [6,7,8,9]. Through protein degradation, these systems are also responsible for modulating cytokine expression, transcription, and epigenetic factors [10,11,12]. Furthermore, cytokines and growth factors modify signaling pathways that promote protein assembly and organelle turnover [10,11,12]. The complexity of the mechanisms that induce muscle atrophy is regulated by non-coding RNAs, including microRNAs (miRNAs).

miRNAs are small non-coding RNAs that control gene expression post-transcriptionally [13,14,15,16,17], leading to global effects on skeletal muscle fibers [18,19,20]. This involves preferential targeting of mRNAs encoding transcription factors, kinases, and phosphatases, leading to amplified impacts [21]. These miRNA-mediated effects orchestrate pathways and biological functions, broadening their spectrum of action in skeletal muscle function and diseases.

Among the diverse miRNAs acting on skeletal muscles, miR-155 plays a crucial role in regulating the immune system, aging-related alterations, development, regeneration, and muscle wasting in cancer-associated cachexia [22,23,24,25,26,27,28,29]. miR-155 influences myoblast proliferation and differentiation into myotubes during in vitro myogenesis [23,27] and is consistently increased in primary muscular disorders, such as Duchenne muscular dystrophy (DMD) [22,26]. Under such conditions, Eisenberg et al. [22] demonstrated that mRNA–miRNAs predicted interactions in DMD that participate in muscle regeneration, suggesting a specific physiological pathway underlying disease pathology.

Given the emerging cooperative property of molecular networks regulated by miRNA targets, identifying the global transcriptional modulation triggered by miR-155 can help us understand post-transcriptional mechanisms in muscle diseases. Here, we characterized the transcriptional profile of muscle cells in response to increased miR-155 expression to identify direct and indirect sets of genes involved in skeletal muscle atrophy. We used computational biology and in vitro approaches to identify potential transcripts regulated by miR-155. Our investigation involved the C2C12 muscle cells and skeletal muscle samples obtained from individuals with DMD.

## 2. Results

### 2.1. Relevance of miR-155 in Different Skeletal Muscle Conditions

We reviewed the literature and observed that the expression of miR-155 is altered in different myopathies, muscular dystrophies, muscle regeneration, and embryonic development of skeletal muscles (Table 1). We highlight the Eisenberg study [22], which analyzed a wide age range of male and female participants, from newborns to 79 years, primarily focusing on the quadriceps and biceps muscles. In this study, the authors noted a differential expression of miR-155 in muscle samples from patients with Duchenne muscular dystrophy, facioscapulohumeral muscular dystrophy, limb–girdle muscular dystrophies R1 and R2, Miyoshi myopathy, nemaline myopathy, polymyositis, dermatomyositis, and inclusion body myositis. In addition to alterations in miR-155 expression in primary skeletal muscle diseases [22], we performed a meta-analysis of transcriptome data obtained from different experimental murine or human samples with altered expression of miR-155 in skeletal muscle or C2C12 cells (Appendix A).

### 2.2. miR-155 Induces a Transcriptional Profile Associated with Morphological Changes

To assess the potential changes in the myotube area induced by the increased expression of miR-155, we transfected C2C12 myotubes with mimic-miR-155-5p. This analysis revealed that miR-155 transfection significantly reduced the number and area of multinucleated myotubes (Figure 1A,B). This fact suggests that miR-155 may interfere with the terminal stages of myogenesis, reducing the number of mature myotubes. Since it was previously demonstrated that miR-155 overexpression reduces myoblast proliferation and migration [23], we sought to evaluate the gene expression profile in myoblasts and myotubes treated with mimic-miR-155. Our transcriptome analysis using RNA-Seq of C2C12 cells transfected with miR-155 revealed 215 dysregulated transcripts in myoblasts (109 upregulated and 106 downregulated) and 359 in myotubes (165 upregulated and 194 downregulated) (Figure 1C,D, Appendix A). Additionally, the overlapping analysis showed that miR-155 overexpression decreased the expression of *Ramp3* in both myoblasts and myotubes (*p*-value: 0.392; Figure 1E). On the other hand, the overlapping analysis found that miR-155 overexpression increased the expression of *Gm13454*, *Mybpc1*, *Gm16529*, and *Tmem262* in myoblasts and myotubes (*p*-value: 0.001; Figure 1E). Using the miRWalk, miRTarBase, and TargetScan algorithms, we identified 511 transcripts predicted to be direct targets of miR-155. Among these direct targets, we identified five deregulated transcripts (*Cpm*, *Plce1*, *Dync1i1*, *Btc*, and *Nr1h3*) in the transcriptome of C2C12 myotubes transfected with miR-155 (Figure 1F). To the best of our knowledge, this study represents the first transcriptome analysis in muscle cells following miR-155 treatment.

To delve further into the transcriptomic data, we explored potential transcription factors and kinases associated with the indirect targets of miR-155. The transcription factors E2F4, FOXM1, EZH2, and SUZ12 were suggested to regulate genes associated with phosphorylation during cell proliferation and differentiation, sarcomere rupture, and apoptosis (Figure 2). The identified kinases involved in the cell cycle progression, inflammation, and fibrogenesis are shown in Appendix A.

### 2.3. Biological Processes Enriched in the Transcriptome of miR-155-Treated Myoblasts and Myotubes

We also investigated the biological processes enriched by DEGs in myotubes and myoblasts treated with miRNA-155 (Figure 3). Overexpression of miR-155 induced specific transcriptional changes in C2C12 myoblasts and myotubes. The upregulated genes in miR-155-treated myoblasts were related to sarcomere organization, increased inflammatory responses, interleukin-6-mediated signaling pathways, and macrophage activation (Figure 3). In myotubes, genes with increased expression enriched cell cycle processes such as the microtubule cytoskeleton organization involved in mitosis, a complex-dependent catabolic process promoting anaphase, and nucleus division (Figure 3). Analysis of downregulated genes in myoblasts identified different enriched categories associated with actin filament network formation, extracellular matrix assembly, cell–cell adhesion, and skeletal system development. Furthermore, the downregulated genes were enriched in functions related to protein tyrosine kinase activity, tube diameter regulation, and downregulation of apoptotic cell removal in myotubes (Figure 3). 

### 2.4. The Transcriptional Overlap between miR-155 Target and DMD Patients

Through a study by Eisenberg et al., 2007 [22], we sought to reanalyze the expression of miR-155 as one of the deregulated miRNAs in nine human muscle disorders. The selection criterion for the dataset was to search for homogeneous data between the number (*n*) of healthy controls and patients with muscle disorders. We retrieved a dataset of DMD samples (accession GSE1004) to investigate the differentially expressed genes in dystrophin-deficient patients and healthy skeletal muscles. We sought to identify a transcriptional profile overlapping the expression of miR-155-target genes in treated C2C12 myotubes with mimic-miR-155 and the transcriptome of muscle samples from DMD patients. Gene expression levels in DMD biopsies and normal skeletal muscle (GSE1004) were used for transcriptomic profile analysis. We compared our list of DEGs affected by the overexpression of the miR-155 with DEGs from 12 muscle biopsies with DMD (GSE1004). Overlap analysis identified 20 shared targets (expected number of overlapping genes: 15; *p*-value: 0.169; Figure 4A). Among the direct targets of miR-155, we found three common genes downregulated in dystrophic samples: *Plce1*, *Dync1i1*, and *Nr1h3*, all exhibiting the same differential expression pattern. Of these 20 common targets, we focused on 18 genes that consistently appeared in our data analysis and previous literature (Appendix A) [34,35,36,37,38]. Correlation analysis identified eighteen differentially expressed genes (Figure 4B). Enrichment analysis demonstrated that 20 shared DEGs play a role in biological processes, such as metabolism, cell cycle, muscle cell maintenance, and the immune system. The Circos plot shows the four main categories of biological processes and their respective genes (Figure 4C). Interestingly, we observed that six genes (*Amhr2*, *Ccl3*, *Bub1b*, *Myh6*, *Slc16a4*, and *Trpc1*) are involved in immune responses. Furthermore, to determine which type of immune cell would be the sensor for the immune response in DMD samples, we performed a CIBERSORT analysis that indicated a significant increase in M2 macrophages (Figure 4D,E).

### 2.5. Direct and Indirect Targets of miR-155-Based Network

Analysis of gene pathways under the post-transcriptional control of miR-155 revealed multiple gene interactions contributing to the cellular immune response (Figure 5A). These findings indicate that miR-155 plays a crucial role in regulating inflammatory processes in skeletal muscle under atrophic and dystrophic conditions. We observed that genes indirectly associated with the regulatory network are predominantly translated into kinases and transcription factors. 

Specifically, *CSNK2A1*, which encodes CK2, a constitutively active protein kinase, was identified in this study [39,40,41,42]. Deletion of CK2β in myofibers results in a myasthenic phenotype, whereas CK2α’-null mice exhibit a reduced regeneration area in muscle fibers after injury [42]. CK2 subunits are critical in regulating Myod1 expression and controlling myoblast fusion [42,43]. Additionally, CK2 binds to the tyrosine kinase BUB1B, producing the BUBR1 protein [44]. Appropriate centrosomal localization of BUB1B is paramount for precise chromosome segregation during mitosis and the preservation of genomic stability (Figure 5B). BUBR1 deficiency in skeletal muscle triggers the activation of p14ARF, and this regulation offers protection against aging-related deterioration and cellular senescence [44]. BUBR1 is also involved in mitotic checkpoints and has angiogenic functions [44]. Furthermore, these two essential kinases are regulated by the transcription factor TRIM28, which regulates skeletal muscle size and function [45]. Notably, TRIM28 acts as an indirect target of miR-155, mediating two gene hubs involved in cell cycle processes and muscle cells, thus contributing to the overall dysregulation observed in dystrophic muscles [45] TRIM28 exhibits predominant nuclear localization within skeletal muscle cells, signifying its pivotal role as a crucial constituent in the assembly of regulatory complexes and modulation of specific transcription factors and kinases [45] (Figure 5B).

## 3. Discussion

The present study aimed to identify direct and indirect targets of miR-155 in C2C12 skeletal muscle cells. Furthermore, integrating transcriptome data from individuals with DMD (public data) allowed us to establish crucial connections with our research objectives. Our results showed that miR-155 affected the gene expression profile of myoblasts and myotubes differently. The treatment of C2C12 myotubes with miR-155 regulated the expression of 359 genes mainly associated with inflammatory processes, dysregulation of the cell cycle, and apoptosis. Among these, 20 genes appeared to play pivotal roles in the muscles of DMD patients. Furthermore, the integrative analysis revealed that *Plce1*, *Dync1i1*, *Ramp3*, *Scl16a4*, *Nr1h3*, and *Bub1b* were downregulated in both miR-155-treated myotubes and skeletal muscle of dystrophic patients, whereas *Aldh1l* and *Nek2* were upregulated. Thus, our results reveal a specific set of miR155-target genes potentially involved in the pathophysiology of DMD. 

Several studies have demonstrated that miR-155 is a critical regulator of skeletal muscle plasticity [23,25,27,46]. Our morphometric analysis of myotubes transfected with miR-155 mimetic molecules agrees with these previous studies, demonstrating that miR-155 impairs C2C12 myotubes differentiation [23,27]. We observed that the overexpression of miR-155 significantly reduced the number and area of C2C12 multinucleated myotubes. Our integrative analysis of different studies and datasets identified the relevance of miR-155 in various experimental and clinical conditions that affect skeletal muscles. In addition, when evaluating the transcriptome of C2C12 muscle cells transfected with miR-155, we noted that it mimicked an inflammatory state and compromised myofiber regeneration (Figure 1A). Eisenberg et al., 2007 [22], investigated the expression profile of 185 miRNAs in 10 major muscle disorders in humans, including DMD. The authors observed that among the miRNAs analyzed, miR-155 was dysregulated in nine of these ten primary muscular dystrophies, suggesting the relevance of this miRNA in primary muscle disorders. 

On the other hand, it is known that this miRNA plays an essential role in immune-mediated inflammatory myopathies. Macrophages also act as crucial regulators of the inflammatory response during skeletal muscle regeneration, affecting resident muscle cells, including myogenic and endothelial cells, as well as fibro-adipogenic progenitors involved in fibrofatty scar formation [47]. While macrophage function is tightly coordinated during muscle regeneration, its dysregulation in muscular dystrophies leads to a chronic inflammatory state [47]. Consistent with previous studies, our findings support the notion that dysregulation of miRNAs, including miR-155, occurs in response to inflammation associated with autoimmunity, potentially influencing muscle activation or degeneration processes and implicating muscle cell differentiation in macrophage-mediated inflammatory responses [32,46,48]. Furthermore, miR-155 in the immune response is essential for myeloid cell activation and balanced regulation of M1 and M2 macrophages during muscle regeneration [25]. Our results showed an increase in M2 macrophages, as estimated by cell type abundance. This may seem like a confirmation bias, given that M2 accumulation is a known consequence of tissue remodeling in DMD. However, this might also be interpreted positively, as it involves tissue repair processes. This brings us to an unexplored aspect in our manuscript: the potentially divergent roles of miR-155 in different cell types. miR-155 might exert opposing effects in muscle versus immune cells, which could be detrimental to muscle fibers while beneficial for immune functions. Our reanalysis of the study by Meyer and Lieber et al., 2012 [49], in which desmin was deleted in mice, resulted in skeletal muscle fibrosis and a significant increase in miR-155 expression (Appendix A). Moreover, infectious processes associated with pathogens and inflammatory stimuli, such as TNF or interferons, and even injury processes, lead to a rapid increase in the expression of miR-155 [50,51]. Considering that fibrotic muscle adaptation without desmin increases the number of inflammatory cells, we note that myoblasts treated with mimic-miR-155 corroborate these investigations [52]. Although our findings and previous studies have demonstrated that this miRNA impairs myotube differentiation, the underlying molecular mechanisms driving its expression remain unknown.

Our transcriptome analysis indicated that miR-155 directly or indirectly controls genes that regulate biological functions in skeletal muscle diseases. Myoblasts and myotubes exhibited different gene expression patterns following miR-155 treatment. Quantitatively, myotubes had 144 additional DEGs compared with myoblasts. Regarding the direction of expression, only Receptor Activity-Modifying Protein 3 (Ramp3) was downregulated in myoblasts and myotubes. In addition, only four transcripts were common when comparing upregulated genes in myoblasts and myotubes (*Gm13464*, *Mybpc1*, *Gm16259*, and *Tmem262*). Given the role of miRNAs as critical regulators of myogenesis, our observations revealed distinct gene expression profiles between myoblast and myotube stages, highlighting the enrichment of different gene sets (Figure 3). Specifically, in the myoblast stage, the enriched processes were primarily associated with differentiation and immune regulation, whereas in myotubes, the enriched processes were predominantly related to the cell cycle. Among the 359 dysregulated genes in C2C12 myotubes, 5 are potential direct targets of miR-155: *Cpm*, *Plce1*, *Dync1i1*, *Btc*, and *Nr1h3* (Figure 4A). Among these targets, *Cpm* showed an increase in expression in C2C12 myotubes with mimic-miR-155, which corroborates the findings of previous studies [53,54,55], linking this upregulation to inflammation, monocyte-to-macrophage differentiation, and M2 macrophage maturation [53,54,55]. *Cpm* encodes a phosphoinositol-linked endopeptidase, an enzyme also associated with monocyte–macrophage differentiation in human cells of hematopoietic origin, suggesting an association between increased *Cpm* expression and cytotoxic macrophages [53]. Additionally, this gene is involved in macrophage maturation, and its upregulation has been detected as a crucial selective marker for the differentiation of active lipid-laden macrophages, including the appearance of foam cells in vivo [53,54,55]. However, to our knowledge, no studies have shown the interaction between miR-155 and *Cpm* in conditions that induce skeletal muscle alterations. Among the direct targets that showed decreased expression after transfections with miR-155, *Btc* was involved in an angiogenic activity in trials with mice after acute mechanical trauma to the skeletal muscle [36], an essential process during muscle regeneration. Another target, *Nr1h3* (or *Lxrα*), which plays a crucial role in the macrophage response to intracellular bacterial infections [56], inflammatory response, and metabolic homeostasis [57,58], showed the same pattern of expression, as observed in DMD samples. Although the ability of *Nr1h3* to regulate its promoter induces the physiological response of macrophages to lipid loading, its expression in mouse cells or tissues is not similarly detectable. Still, the basis for this interspecies difference is unknown [57,59].

To understand how miR-155-affected genes might be involved in muscular dystrophies, we reanalyzed the transcriptome of DMD patients and found 20 overlapping genes with our data. Some of these genes play essential roles in immune response [37,38] (Appendix A). In myotubes, inhibiting the *Aldh1l1* transcription pathway restores oxidative stress and causes mitochondrial dysfunction [35], *Tcap* knockdown inhibits the differentiation of myoblasts into myotubes [34,60], and its null mutation causes muscular limb–girdle muscular dystrophies R2 (LGMDR2) [61]. *Bub1b* encodes a protein associated with mitotic checkpoint control [62], and *Ccl3* is a critical chemokine for idiopathic inflammatory myopathies, with high expression in injured skeletal muscle and responsible for recruiting Treg cells to these sites [38,63]. Altogether, these data suggest that miR-155 and its targets *Plce1*, *Dync1i1*, *Ramp3*, *Scl16a4*, *Nr1h3*, *Bub1b*, *Aldh1l1*, and *Nek2* are involved in the pathological immune response of muscles in DMD patients, as indicated by the overlapping genes that control the inflammatory response. DMD samples showed a higher proportion of M2 macrophages than in healthy human muscle tissues, and we reckoned that the miR-155 could be delivered to the muscle via M2 macrophages.

Moreover, the interaction with the set of genes must be validated in future studies to verify that miR-155 affects the phenotype of dystrophic cells. In addition, it is worth mentioning that the results showing an essential change in the immune system corroborate studies of the top 18 genes shared between DMD and C2C12 myotubes with mimic-miR-155. Interactions with this set of genes should be validated in future studies to verify whether miR-155 affects the dystrophic cell phenotypes. 

## 4. Materials and Methods

### 4.1. Cell Culture and Muscle Differentiation

C2C12 myoblast cells (ATCC^®^ CRL-1772 TM) were cultured in Dulbecco’s modified Eagle’s Medium (DMEM, Thermo Fisher Scientific, Waltham, MA, USA) supplemented with 10% Fetal Bovine Serum (FBS, Thermo Fisher Scientific, Waltham, MA, USA), 1% penicillin–streptomycin (Thermo Fisher Scientific, USA) in a humidified incubator at 37 °C and 5% CO_2_. Subsequently, the cells were transferred to and cultured in 6-well plates (1 × 10^5^ cells/well). C2C12 myoblasts were collected or induced to undergo myogenic differentiation after transfections with mimic-miR-155-5p. For differentiation, once the myoblasts reached 80–90% confluence, the growth medium was replaced with FBS-free, DMEM-containing Horse Serum (2%), L-glutamine, and penicillin/streptomycin (1%) for 120 h. All experiments were performed in triplicate for each group.

### 4.2. Oligonucleotides and Transfection

The mimic-miR-155-5p (mirVanaTM miRNA Mimic, code: 4464066; MC13058—MC10203, Thermo Fisher, Waltham, MA, USA) and the respective negative control (CT) (mirVanaTM miRNA Mimic Negative Control, code: 4464058, Thermo Fisher, Waltham, MA, USA) were transfected into C2C12 myoblasts, at 80% confluence. For transfection, the two complexes were combined into the final transfection solution. First, Lipofectamine RNAiMAX (Thermo Fisher, Waltham, MA, USA) was diluted in Opti-MEM^®^ Reduced Serum Medium (Thermo Fisher, Waltham, MA, USA) to form the first complex. Next, mimic-miR-155 oligonucleotides and negative control were diluted in Opti-MEM^®^ to form the second complex. Finally, the Lipofectamine + Opti-MEM^®^ complex was mixed with the oligonucleotide + Opti-MEM^®^ complex and incubated for 5 min at room temperature. After this period, 250 μL of the final transfection solution was added to each well containing C2C12 cells (80% confluent) in a normal growth medium. The cells were then incubated for 15 h. Next, the myoblasts were transferred to a medium containing 2% horse serum to induce differentiation in myotubes. The myotube area and gene expression were analyzed after five days of differentiation. The experimental design for functional analysis of the miR-155-mimic during myogenesis and the miR-155 expression is described in Appendix A.

### 4.3. Immunostaining

For immunostaining, C2C12 myotubes treated in 6-well plates were fixed with 4% paraformaldehyde for 15 min, washed with PBS and 0.1% Triton X-100 (Sigma, St. Louis, MO, USA), and incubated with a blocking solution containing 1% glycine, 3% BSA, 8% SFB in PBS and Triton X-100 for 1 h at room temperature. Primary antibody (Myh2) was incubated at 1:600 dilution, then overnight at 4 °C, washed with PBS, incubated with secondary antibody (anti-rabbit) at 1:600 dilution for 2 h at 4 °C, and counterstained with DAPI. Digital fluorescent images were captured at room temperature using a TCS SP5 confocal scanning microscope (Leica Microsystems, Wetzlar, Germany). Myh2 pixels were counted using TCS SP5 (Leica Microsystems, Wetzlar, Germany). ImageJ software measured the total nuclei, myotube nuclei, and myotube area. The fusion index was determined as (total myotube nucleus/total nucleus) × 100.

### 4.4. Total RNA Extraction

According to the manufacturer’s instructions, total RNA was extracted from C2C12 myotubes using the TRIzol reagent (Thermo Fisher Scientific, USA). Total RNA was quantified by spectrophotometry using a NanoVue spectrophotometer (GE Life Sciences, Chicago, IL, USA). The extracted RNA was treated with TURBO DNase (Thermo Fisher Scientific, USA) to remove contamination with genomic DNA. RNA quality was determined by RNA Integrity Number (RIN) using a 2100 Bioanalyzer system (Agilent Technologies, Santa Clara, CA, USA). Samples with RIN > 9 were considered in the subsequent analysis.

### 4.5. RNA Sequencing

The construction of the RNA-Seq library for the CT (*n* = 3) and Mimic-mir-155 (*n* = 3) groups was based on 5 μg of total RNA according to the manufacturer’s protocol using the Illumina HiScanSQ Instrument (Illumina, San Diego, CA, USA) and sequenced in the same flow cell as paired-end (2 × 100 bp). The sequencing generated an average of 25 million paired-end readings per sample. Raw sequence reads (.fastq files) were subjected to quality control analysis using the FastQC tool (version 0.11.5, http://www.bioinformatics.babraham.ac.uk/projects/fastqc/, accessed on 30 May 2018), and the process considered average quality scores Phred 20. For reading mapping of cDNA fragments, we used TopHat (version 1.3.2, http://tophat.cbcb.umd.edu, accessed on 30 May 2018) [64], a reading mapping algorithm capable of aligning RNA-Seq readings to a reference transcriptome in the case of the mouse (RefSeq, mm10). To count transcribed mapped readings and perform differential expression analysis, the R software HTSeq and DEseq packages (version 4.1.2, https://cran.r-project.org/bin/windows/base/, accessed on 30 May 2018), respectively, were used. The normalized count tables from myoblasts and myotubes can be accessed in Appendix A. We considered a fold change of 1.5 and a *p*-value < 0.05.

### 4.6. Expression Pattern Visualization 

The transcriptome data for C2C12 myoblast and myotube cells were represented by heatmaps generated using Morpheus online software (https://software.broadinstitute.org/morpheus, accessed on 29 November 2019), which allows easy visualization and matrix analysis of the datasets. Venn’s graphs, which showed the interaction of the myoblasts’ and myotubes’ DEGs, were generated from Cacoo by Nulab 2019 software (https://cacoo.com/, accessed on 22 March 2019). 

### 4.7. Prediction of Direct Target Transcripts of miR-155

Potential miR-155 targets were predicted using the computational algorithms miRWalk [65], miRTarBase [66], and TargetScan [67]. Using more than one algorithm becomes essential to expand the number of predicted targets and filter the search by considering those mRNAs predicted by at least four distinct algorithms as possible targets. MiRWalk v.3.0 (http://mirwalk.umm.uni-heidelberg.de/, accessed on 30 November 2018) provides experimentally predicted and validated information (miRNA) about the miRNA–target interaction. This algorithm offers target miRNA predictions within the complete sequence for humans, rats, and mice. MiRTarBase v.7.0 (https://mirtarbase.cuhk.edu.cn/~miRTarBase/miRTarBase_2022/php/index.php, accessed on 30 November 2018) has over 360,000 microRNA–target interactions. The miRNA–target interactions collected are experimentally validated by reporter assays, Western blotting, microarray, and sequencing experiments. TargetScan v.7.2 (http://www.targetscan.org/vert_72/, accessed on 30 November 2018) predicts miRNA biological targets by scanning for the presence of conserved 8mer, 7mer, and 6mer sites that correspond to regions essential for miRNA binding in the mRNA and uses curated updated miRNA families from Chiang et al., 2010 [68] and Fromm et al., 2015 [69].

### 4.8. In Silico Prediction of Transcriptional Factors and Kinases

A network of transcriptional factors and kinases was predicted to regulate the differentially expressed genes of miR-155-treated myotubes, that is, the potential indirect targets, using the computational algorithms of the eXpression2Kinases [70] (X2K Web; kinases, and transcriptional factors), and STRING Consortium v.11.0 [71] (protein–protein interaction). First, we considered transcriptional factors and kinases predicted by the X2K Web (http://amp.pharm.mssm.edu/X2K/, accessed on 30 November 2018), with a *p*-values < 0.05. We then compared differentially expressed genes to the list of genes translated into mouse kinases and transcription factors to identify which presented altered expression. Finally, we connected these transcription factors and enriched kinases through known protein–protein interactions (PPIs) to build a subnetwork. The categories with a *p*-value < 0.05 were considered statistically significant.

### 4.9. Enrichment Analysis

DEGs were used to identify enriched biological processes using the EnrichR tool [72,73], powered by the Gene Ontology Consortium [74] (The Gene Ontology Consortium, 2019) (http://geneontology.org/, accessed on 29 March 2019) library ‘GO_Biological_Process_2018’, by PANTHER version 17.0 [75] (available at http://www.pantherdb.org/, accessed on 29 March 2019). Gene ontology (GO) categories with a *p*-value < 0.05 were statistically significant. We used the REViGO [76] tool (http://revigo.irb.hr/, accessed on 29 March 2019) to summarize long lists of GO terms by removing redundant gene ontology terms. Ontology data were plotted using GraphPad Prism 8 software (https://www.graphpad.com/, accessed on 29 March 2019). 

### 4.10. Differential Expression Analysis of Dystrophin-Deficient Patients

We retrieved a dataset of DMD samples (accession GSE1004) to investigate the differentially expressed genes in the skeletal muscles of dystrophin-deficient patients and healthy individuals. The dataset used in this analysis was selected from the GEO public repository maintained by the National Center for Biotechnology Information (NCBI) (https://www.ncbi.nlm.nih.gov/geo/, accessed on 20 December 2019) [77]. The intensity table was downloaded and processed, and DEGs between groups were identified using the Limma-Voom pipeline of the GEO2R web tool (https://www.ncbi.nlm.nih.gov/geo/geo2r/, accessed on 20 December 2019). Thus, the log transformation was automatically applied to the data using GEO2R. We applied the statistical cut-offs of log2 fold change > 1.2 and *p*-value < 0.05 to determine DEGs between DMD and normal samples. Next, we compared our list of 359 differentially expressed genes in miR-155-treated myotubes. The CIBERSORTx tool (https://cibersortx.stanford.edu/, accessed on 20 December 2019) was used to estimate cell fractions by relative proportion in the reanalysis of the two groups previously compared in GEO2R: (1) healthy skeletal muscle samples (n = 11) and (2) dystrophin-deficient patients (n = 12) [78]. Gene expression normalized data with standard annotation was loaded into the CIBERSORTx algorithm, processed using the LM22 signature and 1000 permutations, and we considered fractions with *p*-values < 0.05. Genes in major functional categories of the top genes shared between DMD samples and C2C12 myotubes with mimic-miR-155 were displayed using the Circos plot (http://circos.ca/, accessed on 20 December 2019) [79].

### 4.11. Reconstruction of Molecular Networks and Data Visualization

The direct targets of miR-155, in addition to its transcriptional factors (TFs) and predicted kinases (see in “In silico prediction of transcriptional factors and kinases” section), were grouped by overlapping genes that were also deregulated in DMD samples. PPI networks were generated using the STRING Consortium v.11.0 [71]. All interactions were derived from laboratory experiments with high-performance screening, text mining, and previous knowledge in selected databases with a high confidence level (sources: experiments, databases; confidence score ≥ 0.90). Furthermore, visualization and annotation of data from gene–PPI interaction networks were performed using the Cytoscape tool [80]. Finally, the graphical representation of the miR targets inside the muscle cell was created with BioRender.com.

### 4.12. Literature Review and Meta-Analysis 

We performed a meta-analysis of studies available in the literature (PubMed; https://www.ncbi.nlm.nih.gov/pubmed) that have identified changes in miR-155 expression in skeletal muscle or C2C12 cells under different experimental conditions. In addition, we searched the Entrez GEO Profiles database (https://www.ncbi.nlm.nih.gov/geoprofiles/, accessed on 20 December 2019) using the keywords “*miR-155 and skeletal muscle*”, focusing on clinical studies and cell models (human and murine). We selected ten studies on different conditions, such as aging, muscular dystrophies, physical exercise, and models of skeletal muscle atrophy (Appendix A). These selected studies showed alteration in the expression levels of miR-155 (*p*-value < 0.05; control vs. condition), according to the results of the GEO2R tool (https://www.ncbi.nlm.nih.gov/geo/geo2r/, accessed on 20 December 2019). The reanalyzed miR-155 expression data were presented in a forest plot generated by the Comprehensive Meta-Analysis software V3 (https://www.metaanalysis.com/index.php?cart=BXVZ2967855, accessed on 20 December 2019).

### 4.13. Statistical Analysis

Unless otherwise indicated, values are reported as mean ± standard deviation (SD). The Student’s *t*-test was used to establish the DEGs (GraphPad Prism V.9) with significant values. *p*-values < 0.05 were considered statistically significant.

## 5. Conclusions

In conclusion, our findings indicate that miR-155 induces a distinct transcriptional profile of genes encoding proteins associated with anti-proliferative, pro-apoptotic, and inflammatory functions. Digital cytometry analysis of skeletal muscle samples from DMD patients revealed a potential association between miR-155 and M2 macrophages, suggesting its involvement in tissue remodeling and immune regulation. The increased expression of miR-155 leads to the downregulation of genes involved in apoptotic cell clearance, thereby compromising the efficiency of the apoptosis-signaling pathway. This observation highlights the gene expression pattern and regulatory directionality similarity between mimic-miR-155 and DMD. Furthermore, miR-155 directly controls the expression of at least five critical genes and indirectly influences numerous other genes through post-transcriptional processes. Our findings corroborate that miR-155 impairs myotube differentiation in C2C12 muscle cells through the coordinated regulation of genes associated with inflammation and apoptosis.

## Figures and Tables

**Figure 1 ijms-25-01777-f001:**
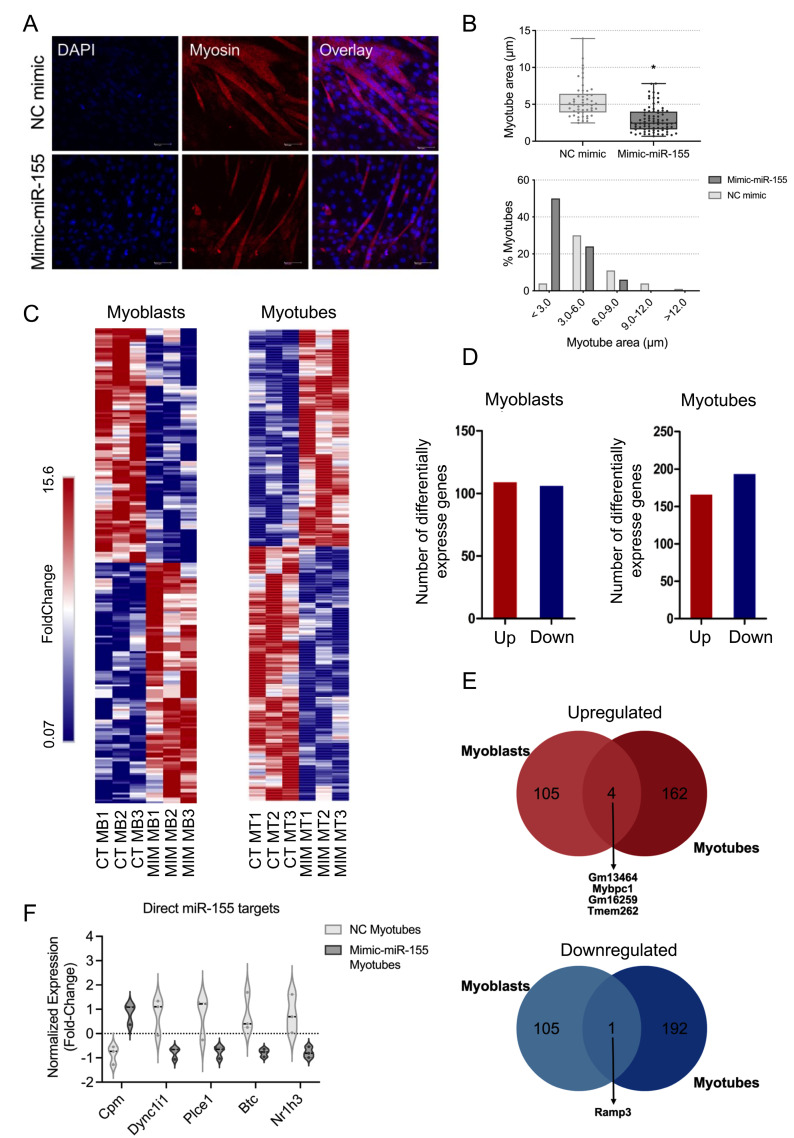
The gene expression profiles of C2C12 myoblasts and myotubes treated with miR-155 are distinct. FC (fold change). (**A**) Immunofluorescence of C2C12 myotubes with mimic-miR-155 stained with an antibody that recognizes Myh2 (myosin heavy chain, red). DAPI-stained nuclei. (**B**) Quantitative analysis of C2C12 myotube size (top) and size distribution (bottom) in the control and miR-155-overexpressing cells. The myofiber area was determined using ImageJ software version 1.52. The data represent the mean ± standard deviation of at least three independent experiments. Statistical significance was analyzed using the Student’s *t*-test. (**C**) Heatmap of differentially expressed genes (DEGs) between myoblast (MB) or myotube (MT) groups overexpressing miR-155 and their respective controls (CT; represent independent biological replicates for each group). Unsupervised hierarchical cluster analysis was performed using DEGs with *p*-values < 0.05 and fold change > 1.5 and is presented as a color scale. (**D**) Bar graphs show the number of DEGs (upregulated and downregulated) in miR-155-treated C2C12 myoblasts (left) and myotubes (right). (**E**) Venn diagram showing DEGs shared between miR-155-treated C2C12 myoblasts and myotubes. (**F**) Normalized expression (RNA-Seq) of five potential direct targets of miR-155, identified by network analysis of C2C12 myotubes treated with miR-155 and their respective controls. *p*-value < 0.05. NC mimic = control. Mimic-miR-155 = treatment. Asterisk (*) indicates *p* value smaller than 0.05 (*p* < 0.05).

**Figure 2 ijms-25-01777-f002:**
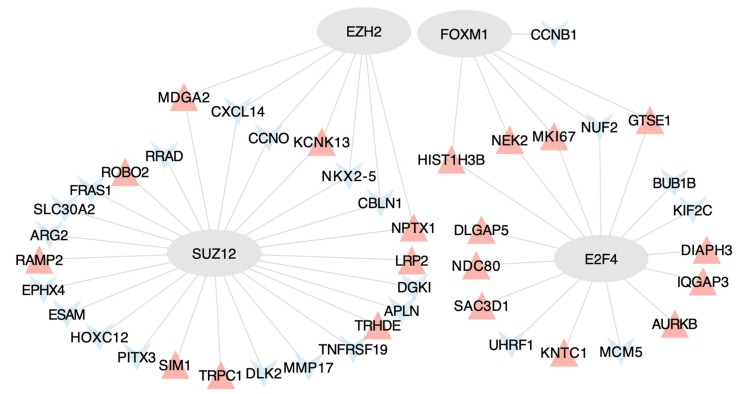
Regulatory network depicting the interaction between transcription factors and genes differentially expressed by miR-155 in muscle cells. Grey ellipses symbolize the transcription factors; blue triangles indicate downregulated genes and red triangles denote upregulated genes.

**Figure 3 ijms-25-01777-f003:**
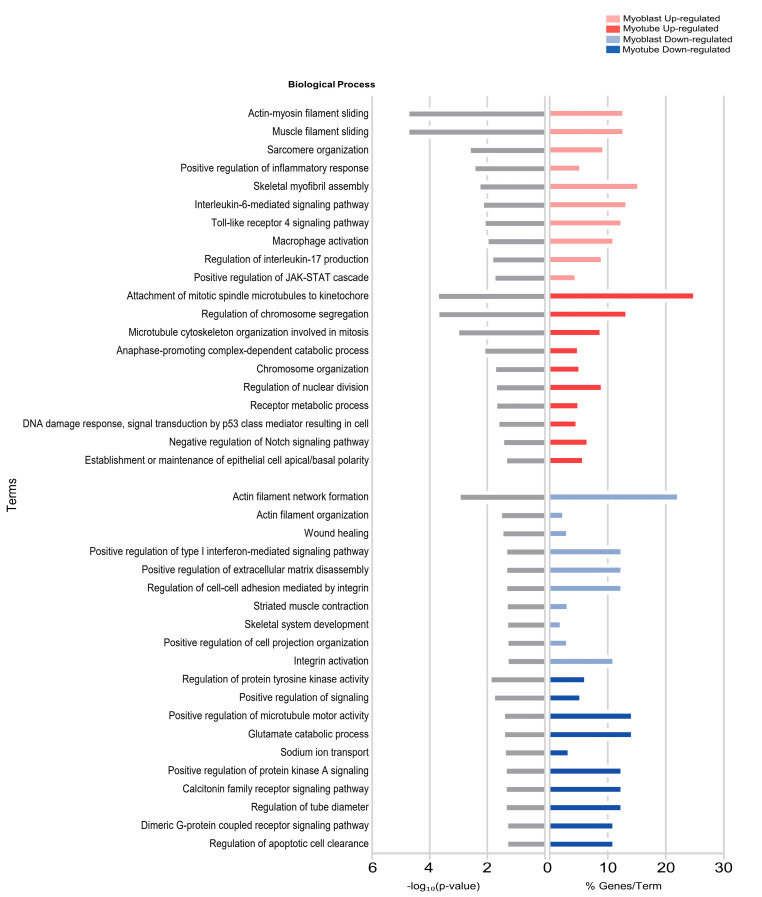
Biological processes enriched by miR-155-treated myoblasts and myotubes. C2C12 myotubes transfected with mimic-miR-155-5p induce gene expression changes associated with different functional categories. Gene ontology of differentially expressed genes (DEGs) in C2C12 myoblasts after treatment with miR-155 to identify critical ontologies. Each horizontal bar on the left (gray bars) represents the number of enriched ontology terms presented in the dataset, considering −log_10_ (*p*-value). Each horizontal bar on the right (colored bars) represents the percentage of genes shown in the dataset compared with the total number of genes in each ontology. Fractions of DEGs in each lane (red, increasing; blue, decreasing) are shown on the x-axis.

**Figure 4 ijms-25-01777-f004:**
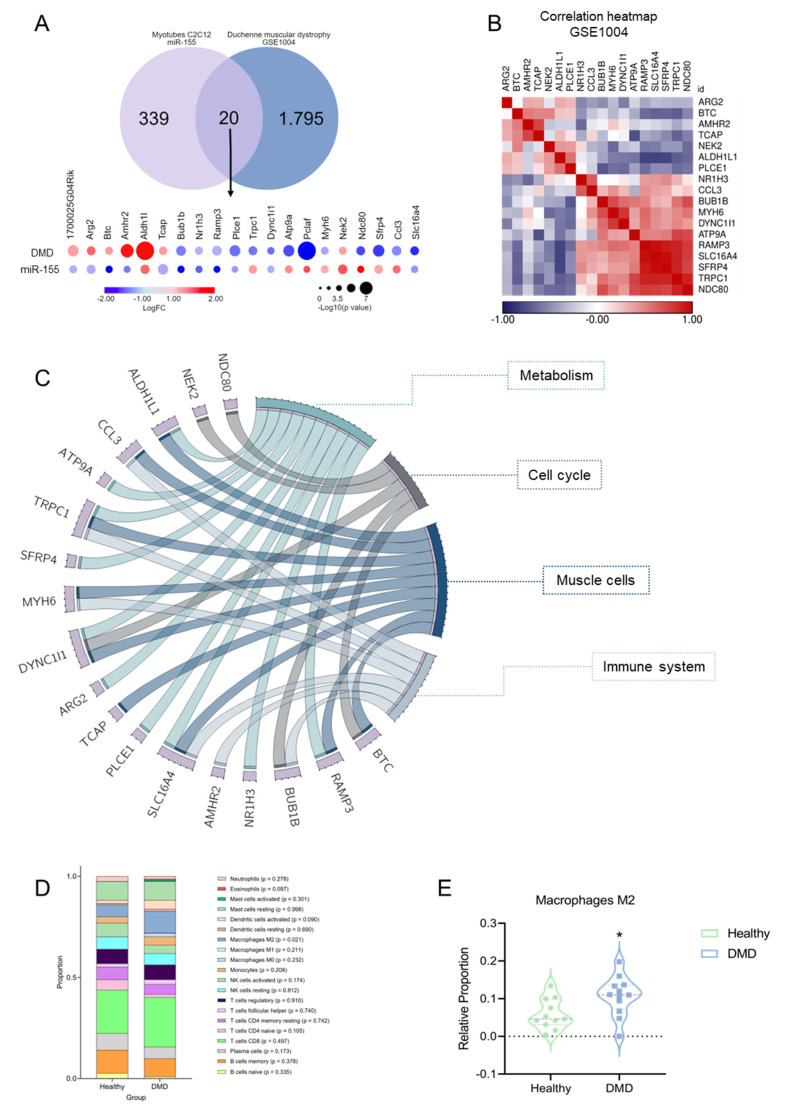
miR-155 mediates immunomodulatory pathology in Duchenne muscular dystrophy (DMD). (**A**) A comparative bubble heatmap plotted in nodes to represent 20 differentially expressed genes in patients with DMD and C2C12 myotubes treated with miR-155. Consider the colors for log_10_FC (fold change) and dimensioned by −log_10_ (*p*-value). (**B**) Correlation heatmap of gene expression data using microarrays (HG_U95Av2) of quadriceps biopsies from 12 DMD patients. Unsupervised hierarchical cluster analysis was performed using DEGs with *p*-values < 0.05 and fold changes > 1.5 and is presented as a color scale. The data are presented in Appendix A. (**C**) The circle represents the genes shared between the four biological categories (metabolism, cell cycle, muscle cells, and immune system). (**D**) The proportion of immune cells associated with differentially expressed genes was similar between patients with DMD and healthy controls. (**E**) The relative proportion of M2 macrophages in 12 muscle samples from DMD patients was statistically significant compared with healthy controls. * *p*-value = 0.021.

**Figure 5 ijms-25-01777-f005:**
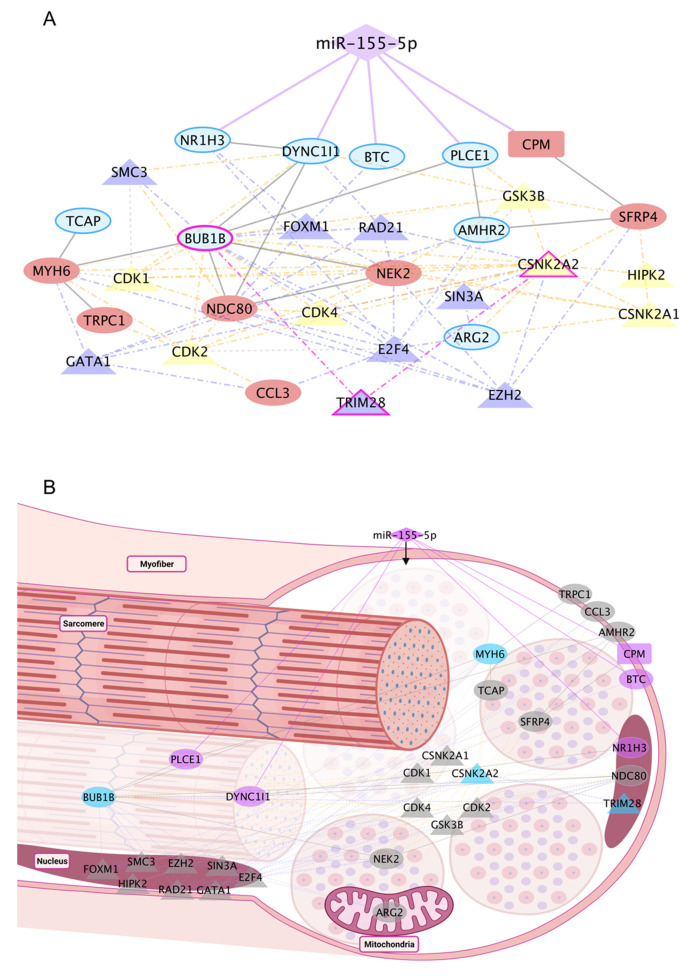
Integrated analysis revealed PPI pathways and regulatory factors associated with increased miR-155 expression in C2C12 skeletal muscle cells and individuals with DMD. The interaction network was predicted by the overlap of direct targets, TFs, kinases, and common targets with DMD (GSE1004). (**A**) Gene regulatory network of differentially expressed genes induced by miR-155 overexpression. The five direct targets are linked to miR-155 by a solid purple line. Purple triangles represent transcription factors, and yellow triangles represent kinases. The genes described in blue show decreased expression in C2C12 cells treated with mimic-miR-155, and the genes shown in red show increased expression. Dashed lines represent indirect interactions caused by transcriptional factors or kinases. A solid gray line represents interactions between PPIs. (**B**) The arrangement of the targets can be visualized in skeletal muscle cells. The TFs are purple, and the kinases are yellow. The solid purple line represents the direct interaction between miR-155-5p and the five direct targets. Purple nodes highlight the direct targets. Genes common to DMD are shown in ellipses. The rectangle highlights the *CPM* gene, which does not overlap with DMD. Diamond-shaped nodes represent TFs and kinases. *BUB1B*, *CSNK2A2*, *TRIM28*, and *MYH6* are highlighted in blue as key targets (DMD and miR155).

**Table 1 ijms-25-01777-t001:** Studies reporting skeletal striated muscle conditions affected by miR-155 expression.

Condition	Clinical or Experimental Study	Ref.
Primary muscular disorders	Duchenne muscular dystrophy, facioscapulohumeral muscular dystrophy, limb–girdle muscular dystrophies R1 and R2, Miyoshi myopathy, nemaline myopathy, polymyositis, dermatomyositis, and inclusion body myositis	[22]
Skeletal muscle differentiation	C2C12 mouse myoblast cells	[27]
Skeletal muscle differentiation	C2C12 mouse myoblast cells	[30]
Myogenesis	Primary human skeletal muscle cells (hSkMC) and murine PMI28 cells	[24]
Skeletal muscle regeneration	Mouse skeletal muscle	[25]
Insulin-sensitive tissues	Mouse skeletal muscle	[31]
Skeletal muscle differentiation	C2C12 mouse myoblast cells	[23]
Skeletal muscle differentiation	Mouse skeletal muscle and C2C12 mouse myoblast cells	[32]
Sarcopenia and type 2 diabetes mellitus	Human muscle samples	[33]

## Data Availability

The RNA-Seq count table is available in Appendix A. The DMD-published microarray data can be found in the GEO database (GSE1004).

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
