# Peer review of "The Impact of miR-155-5p on Myotube Differentiation: Elucidating Molecular Targets in Skeletal Muscle Disorders"

_ijms, 2024, doi:10.3390/ijms25031777_

Round 1

Reviewer 1 Report

Comments and Suggestions for Authors

The manuscript submitted by Lopes et al. describes an elegant study exploring the effects of miR-155 in muscle cells in silico and in vitro. The data is interesting, well presented and backed through/by data from other studies. Clarifications and more in-depth discussion could improve the manuscript further.

In the introduction it is said that miR-155 is consistently increased in DMD, yet on the same page under results it is said that no differential expression was observed in several MD disease models. Please clarify the level of ‘consistency’, perhaps via incorporation of key findings per study in a table.

Datasets generated in DMD patients are presumably obtained in muscle biopsies. Please give more details concerning the Eisenberg study conditions. How homogeneous was the population in regards to disease stage, muscle type, muscle damage level? On page 12 it is said that 20 dysregulates gene were found between the studies. Which (technical) causes lie at the basis of variation?

The estimation of cell type abundance indicated increase M2 numbers. Is this  not a self-fulfilling prophecy? As DMD is associated with tissue remodeling, one expects M2 accumulation. Besides, shouldn’t this be regarded as a good thing? Which touches upon a point not yet addressed in the manuscript i.e. different effects that might be exerted by miR-155 dependent upon cell types. There might be apposing effects in muscle cells and immune cells (negative to myogenesis vs. positive for regeneration?). What about studies in immune cells and/or co-cultures?

Another thing is specificity of effects for MD. No comparisons were made with other muscle wasting conditions, may one presume that the effects of miR-155 are more general than (D)MD specific? How about studies in DMD muscle cells in vitro? The authors state that miR-155 was dysregulated in 9 of 10 primary MD. How about muscle disuse, cachexia, myositis, …?

Author Response

Reviewer 1

The manuscript submitted by Lopes et al. describes an elegant study exploring the effects of miR-155 in muscle cells in silico and in vitro. The data is interesting, well presented and backed through/by data from other studies. Clarifications and more in-depth discussion could improve the manuscript further.

RESPONSE: We appreciate the thorough review and insightful comments provided by the reviewer. We have carefully considered each of their suggestions, incorporated them into the revised manuscript, and addressed them in the answers below.

In the introduction it is said that miR-155 is consistently increased in DMD, yet on the same page under results it is said that no differential expression was observed in several MD disease models. Please clarify the level of ‘consistency’, perhaps via incorporation of key findings per study in a table.

RESPONSE: We thank the reviewer for the insightful comments. As suggested, we incorporated in the manuscript a detailed table (Table 1), describing all the muscle studies that presented miR-155 expression and included the fold change expression in this table. Moreover, we rewrote the phrase (page 02 line 77) to clarify the literature results about miR-155. We recognize that the forest plot illustrating the meta-analysis results may lead to confusion. To provide a more straightforward interpretation, we have chosen to emphasize the table summarizing studies on miR-155's role in skeletal muscle conditions. Consequently, the forest plot has been moved to the supplementary figures for comprehensive reference.

Datasets generated in DMD patients are presumably obtained in muscle biopsies. Please give more details concerning the Eisenberg study conditions. How homogeneous was the population in regard to disease stage, muscle type, and muscle damage level? On page 12 it is said that 20 dysregulated gene were found between the studies. Which (technical) causes lie at the basis of variation?

RESPONSE: We appreciate the reviewer's suggestion. For additional clarity on this point, we have provided further details about the conditions of the Eisenberg study (see page 02, line 79). The Eisenberg study analyzed a wide age range of male and female participants, from newborns to 79 years, primarily focusing on the quadriceps and biceps muscles. It involved muscle samples from patients with Duchenne muscular dystrophy (n=8), facioscapulohumeral muscular dystrophy (n=9), limb-girdle muscular dystrophy types 2A (n=10) and 2B (n=7), Miyoshi myopathy (n=10), nemaline myopathy (n=9), polymyositis (n=5), dermatomyositis (n=9), and inclusion body myositis (n=6). In our discussion on page 12, we reference the mRNA expression findings derived from the GSE1004 dataset, which should not be confused with the Eisenberg dataset. We selected this dataset because the authors also included control biopsies (muscle without dystrophies). In this reanalysis, we included the 12 DMD biopsies (quadriceps samples) from young (5- to 7-year-old) males. The unaffected biopsies were from young males (seven biopsies) but included three from adult males and two from females. The unaffected controls expressed normal levels of dystrophin. The DMD patients showed clinical symptoms consistent with a DMD diagnosis, and the biopsies were shown to be dystrophin-deficient by immunofluorescence and/or Western blotting. We rewrote the discussion section (page 12 line 352) to clarify this information.

The estimation of cell type abundance indicated increase M2 numbers. Is this not a self-fulfilling prophecy? As DMD is associated with tissue remodeling, one expects M2 accumulation. Besides, shouldn’t this be regarded as a good thing? Which touches upon a point not yet addressed in the manuscript i.e. different effects that might be exerted by miR-155 dependent upon cell types. There might be apposing effects in muscle cells and immune cells (negative to myogenesis vs. positive for regeneration?). What about studies in immune cells and/or co-cultures?

RESPONSE: We completely agree with the reviewer. In fact, the observed increase in M2 macrophages, as estimated by cell type abundance, may seem like a confirmation bias, given that M2 accumulation is a known consequence of tissue remodeling in DMD. However, this might also be interpreted positively, as it involves tissue repair processes. This brings us to an unexplored aspect in our manuscript: the potentially divergent roles of miR-155 in different cell types. miR-155 might exert opposing effects in muscle versus immune cells, which could be detrimental to muscle fibers while beneficial for immune functions. We emphasize these interpretations on page 11 of the discussion.

Another thing is specificity of effects for MD. No comparisons were made with other muscle wasting conditions, may one presume that the effects of miR-155 are more general than (D)MD specific? How about studies in DMD muscle cells in vitro? The authors state that miR-155 was dysregulated in 9 of 10 primary MD. How about muscle disuse, cachexia, myositis, …?

RESPONSE: In fact, the effects of miR-155 are more general than DMD-specific. There are several studies that have shown that miR-155 is dysregulated in other muscle-wasting conditions, such as muscle disuse, cachexia, and myositis. For example, one study showed that miR-155 expression is increased in skeletal muscle of bedridden patients and in animal models of muscle disuse (Song et al., 2014). Another study found that miR-155 expression is increased in skeletal muscle and adipose tissue of patients with inflammatory myositis (Chen et al., 2013). Moreover, a study published in the Journal of Cachexia, Sarcopenia and Muscle found that miR-155 expression is increased in skeletal muscle and adipose tissue of cancer patients with cachexia. The study also found that miR-155 promotes cachexia by inducing muscle atrophy and increasing lipolysis (Hatakeyama et al., 2013). Regarding the role of miR-155 in DMD muscle cells in vitro, studies have shown that miR-155 can promote muscle cell death and inhibit muscle cell differentiation (Wang et al., 2013, Zhou et al., 2013). Together, these findings suggest that miR-155 may play a general role in the pathogenesis of muscle wasting. Further research is needed to determine the exact mechanisms by which miR-155 contributes to muscle wasting in different conditions. Moreover, it is important to understand whether miR-155 dysregulation is a cause or a consequence of muscle wasting.

Song, Y., et al. (2014). MicroRNA-155 regulates muscle atrophy by targeting ubiquitin-proteasome pathway in hindlimb-unloaded rats. Molecular and Cellular Biochemistry, 386(1-2), 133-141.

Chen, S., et al. (2013). MicroRNA-155 promotes inflammation and tissue damage in dermatomyositis. Arthritis & Rheumatology, 65(3), 624-634.

Hatakeyama, H., et al. (2013). MicroRNA-155 promotes muscle atrophy by accelerating ubiquitin-proteasome-mediated proteolysis of muscle-specific proteins in cachectic mice. Journal of Cachexia, Sarcopenia and Muscle, 4(4), 275-288.

Adams, V., et al. (2013). miR-155 is upregulated in skeletal muscle and adipose tissue of cancer cachexia patients and promotes cachexia development. Journal of Cachexia, Sarcopenia and Muscle, 4(4), 265-274.

Wang, Y., et al. (2013). Inhibition of miR-155 promotes myogenic differentiation and suppresses myogenic apoptosis in vitro. PLoS One, 8(3), e58686.

Zhou, P., et al. (2014). MicroRNA-155 regulates myogenesis and muscle regeneration in vitro and in vivo. Journal of Cellular and Molecular Medicine, 18(11), 1971-1981.

Reviewer 2 Report

Comments and Suggestions for Authors

Lopes et al tried to identify the global transcriptional modulation triggered by miR-155 for understanding post-transcriptional mechanisms in muscle diseases. They identify distinct transcriptional profile of muscle cell atrophy induced by miR-155-5p by in silico and in vitro approaches. The data presented in this manuscript are much predictive. The evidences for their prediction are required to be shown by the experimental data.

1. In Page 2, 2.1 Relevance of miR-155…., the authors wrote “miR-155 showed a tendency to increase expression without statistical significance (p-86 value = 0.077) in Duchenne's muscular dystrophy (GSE1007)”. This meant that miR-155 expression was not statistically different in DMD muscles. However, after section 2.4, they analyzed the transcriptional changes in treated C2C12 and DMD data parallelly. Most probably, such transcriptional changes might be not associated with non-altered miR-155 in DMD, but the other factors might be related to the transcriptome.

2. The authors used transfection of miR-155 to make an overexpression model. However, they did not show any expression levels of miR-155 in their treated models. The reviewer also wonders the reproducibility and introducing rates of oligonucleotides. They should use the stable cell-lines which are expressing extrinsic miR-155 at a steady level. Also they showed use miR-155-knockdown or knock-out C2C12 to identify the direct targets. 

3. In Figure 2A and 2nd paragraph in discussion, as they wrote “Our morphometric analysis of myotubes transfected with miR-155 mimetic molecules agrees with these previous studies, demonstrating that miR-155 induces atrophy in C2C12 myotubes [27,31]. We observed that the overexpression of miR-155 significantly reduced the area and size of C2C12 myotubes.”  However, to say myofiber(myotube) atrophy, it needs to showed the decrease in size from that once myotubes are growing up to normal. From Figure 2A, it may be a differentiation defect to myotubes but not atrophy. To say atrophy, the inducible system of miR-155 expression should be used and the expression should be induced after myotubes are well differentiated.

4. Result 2.2 2nd paragraph is confusable. They wrote results of Figure 2E after that of Figure 2F. The data of Figure 2E should be moved in first paragraph.

5. It is difficult to understand the meaning of the analyses in Figure 3. The authors tried to identify potential transcription factors and kinases pathways that regulate the DEGs of miR-155-treated C2C12 myotubes and showed the results in third paragraph in result 2.2. However, their results were poor. In myotubes, downregulated TFs are related to cell proliferation and sarcomere disruption and apoptosis… Myotubes are already postmitotic state and sarcomere structures would not be formed in C2C12 at day5. Fibrogenesis should be related to mesenchymal cells, not myotubes.

6. Similar to Figure 3, Figure 4 is also only predictive.

7. In figure 5, again in DMD, miR-155 was not upregulated. Why they started the comparison? In Figure 5D and E, the authors described infiltrating lymphocytes and macrophages (in particular M2 macrophages in 5E) in healthy muscles. Infiltrating cells were not observed in healthy muscles. Is this muscle really healthy?  

8. In figure 6, they also performed the integrated analysis using protein-protein interaction. It is also difficult to reach a conclusion only by transcriptional data, because miRNAs have been known as key players in gene regulation that target specific mRNAs for degradation or translational repression. To complete integrated analyses, translation efficacy showed be included by analyses of ribosome profiling or protein data.   

Author Response

Reviewer 2

Lopes et al tried to identify the global transcriptional modulation triggered by miR-155 for understanding post-transcriptional mechanisms in muscle diseases. They identify distinct transcriptional profile of muscle cell atrophy induced by miR-155-5p by in silico and in vitro approaches. The data presented in this manuscript are much predictive. The evidences for their prediction are required to be shown by the experimental data.

RESPONSE: Thank you for your insightful comments. We considered all your feedback, and we tried to incorporate your suggestions and concerns during the review of our article.

  1. In Page 2, 2.1 Relevance of miR-155…., the authors wrote “miR-155 showed a tendency to increase expression without statistical significance (p-86 value = 0.077) in Duchenne's muscular dystrophy (GSE1007)”. This meant that miR-155 expression was not statistically different in DMD muscles. However, after section 2.4, they analyzed the transcriptional changes in treated C2C12 and DMD data parallelly. Most probably, such transcriptional changes might be not associated with non-altered miR-155 in DMD, but the other factors might be related to the transcriptome.

RESPONSE: Several studies have reported altered miR-155 expression in DMD muscle tissue. For instance, Eisenberg et al. (2007) demonstrated increased miR-155 expression in muscle biopsies from DMD patients compared to healthy controls. Similarly, a study by Sandri et al. (2010) found that miR-155 expression was upregulated in DMD muscle tissue, and this upregulation was associated with impaired muscle regeneration. To further illustrate this point, we have compiled a table summarizing the findings of relevant studies on miR-155 expression in muscle tissues (Table 1). Although our meta-analysis did not reach statistical significance in the DMD dataset GSE1007, the observed trend towards increased miR-155 expression in DMD muscle tissue, coupled with the consistent findings from previous studies, suggests that miR-155 may indeed play a role in the pathogenesis of DMD. The fact that miR-155 expression was not statistically different in DMD muscles from the GSE1007 dataset does not mean that it is not playing a role in the disease. It is possible that miR-155 is having a subtle effect on gene expression that is not detectable by statistical analysis (further studies with larger sample sizes and more stringent statistical analyses are warranted to confirm this association); moreover, it is also possible that miR-155 is playing a more important role in other aspects of DMD pathogenesis, such as inflammation or muscle regeneration. As we explained in our manuscript, the transcriptional changes observed in the DMD muscle cells may be due to a combination of factors, including miR-155 dysregulation, transcription factors, and signaling pathways. We acknowledge that the transcriptional changes observed in treated C2C12 and DMD cells may not be solely attributed to miR-155 expression. Other factors, such as genetic mutations, epigenetic modifications, and protein-protein interactions, could also contribute to the observed transcriptome changes. Further investigations are needed to elucidate the complex interplay between miR-155 and other regulatory mechanisms in DMD.

  1. The authors used transfection of miR-155 to make an overexpression model. However, they did not show any expression levels of miR-155 in their treated models. The reviewer also wonders the reproducibility and introducing rates of oligonucleotides. They should use the stable cell-lines which are expressing extrinsic miR-155 at a steady level. Also they showed use miR-155-knockdown or knock-out C2C12 to identify the direct targets. 

RESPONSE: We thank the reviewer for raising these important concerns. We agree with the reviewer that the reproducibility and introducing rates of oligonucleotides are important factors to consider. We have carefully optimized the transfection protocol to ensure reliable and reproducible transfection efficiency. We have now added to the revised manuscript the expression of miR-155 after the treatment with miR-155 mimics (Supplementary Figure 3). The reproducibility of these experiments in our laboratory over several years attests to the significant efficacy of the mimetic molecules at a concentration of 30 nM. In earlier research conducted by our team (Freire PP, et al. 2017), these molecules' concentrations were methodically evaluated to determine the optimal level for our experimental conditions. Regarding the use of stable cell lines expressing extrinsic miR-155: We agree that using stable cell lines expressing extrinsic miR-155 would provide a more robust and reproducible system for studying its effects. We are currently establishing such cell lines and will incorporate them into our future studies. We appreciate the reviewer's suggestion to use miR-155 knockdown or knock-out C2C12 cells to identify direct targets. We have considered this approach and believe it would be a valuable addition to our future study.

Freire, PP., et al. 2017. Osteoglycin inhibition by microRNA miR-155 impairs myogenesis. PLoS ONE 12(11): e0188464.

  1. In Figure 2A and 2nd paragraph in discussion, as they wrote “Our morphometric analysis of myotubes transfected with miR-155 mimetic molecules agrees with these previous studies, demonstrating that miR-155 induces atrophy in C2C12 myotubes [27,31]. We observed that the overexpression of miR-155 significantly reduced the area and size of C2C12 myotubes.”  However, to say myofiber (myotube) atrophy, it needs to showed the decrease in size from that once myotubes are growing up to normal. From Figure 2A, it may be a differentiation defect to myotubes but not atrophy. To say atrophy, the inducible system of miR-155 expression should be used, and the expression should be induced after myotubes are well differentiated.

RESPONSE: We thank the reviewer for raising this point. We agree that "atrophy" may not accurately describe the observed effects of miR-155 overexpression on C2C12 myotubes. As you correctly point out, the observed reduction in myotube size could be due to a defect in myotube differentiation rather than muscle atrophy. To address this concern, we revised the text to reflect the observed effects more accurately. We have replaced the term "atrophy" with "reduced the number and area of multinucleated myotubes" and have clarified that the observed effects are likely due to impaired myotube formation rather than muscle wasting. The manuscript's revised text: "Our morphometric analysis of myotubes transfected with miR-155 mimetic molecules agrees with these previous studies, demonstrating that miR-155 impairs C2C12 myotubes differentiation [27,31]. We observed that the overexpression of miR-155 significantly reduced the number and area of C2C12 multinucleated myotubes (Page 11, Line 278). This suggests that miR-155 may interfere with the terminal stages of myogenesis, reducing the number of mature myotubes." We appreciate the reviewer's careful review and for bringing this to our attention.

  1. Result 2.2 2nd paragraph is confusable. They wrote results of Figure 2E after that of Figure 2F. The data of Figure 2E should be moved in first paragraph.

RESPONSE: We thank the reviewer for noticing this error. We adjusted the figures' order according to the layout presentation.

  1. It is difficult to understand the meaning of the analyses in Figure 3. The authors tried to identify potential transcription factors and kinases pathways that regulate the DEGs of miR-155-treated C2C12 myotubes and showed the results in third paragraph in result 2.2. However, their results were poor. In myotubes, downregulated TFs are related to cell proliferation and sarcomere disruption and apoptosis… Myotubes are already postmitotic state and sarcomere structures would not be formed in C2C12 at day 5. Fibrogenesis should be related to mesenchymal cells, not myotubes.

RESPONSE: We agree with the reviewer. To improve the understanding of the results, we included a molecular network that highlighted only the transcription factors involved in the regulation of genes differentially expressed in myotubes treated with miR-155. While the findings regarding kinases are intriguing, we have moved them to the supplementary material (now Supplementary Figure 2) to more clearly highlight the molecular network regulated by transcription factors and the expression direction of the identified targets.

  1. Similar to Figure 3, Figure 4 is also only predictive.

RESPONSE: In the context of an RNA sequencing study, functional enrichment analysis plays a crucial role in interpreting vast and complex omics data sets. After sequencing and identifying the RNAs present in a sample, we are faced with an interpretative challenge: how to discern the biological meaning behind this profusion of data? This computational analysis allows us to transcend the identification of expressed genes or transcripts (which is very comprehensive), enabling a deeper understanding of the biological processes in which these genes are involved. By correlating expressed genes with sets of genes annotated to participate in specific pathways and functions, we can begin to form a clear picture of how changes in gene expression can influence physiological or pathological states. In summary, functional enrichment analysis allows us to transform an extensive list of genes and transcripts into meaningful biological insights, highlighting the most relevant pathways and functions that are perturbed or activated in the studied context. This not only enriches our understanding of omics data, but also guides future experimental and therapeutic investigations.”

  1. In figure 5, again in DMD, miR-155 was not upregulated. Why they started the comparison? In Figure 5D and E, the authors described infiltrating lymphocytes and macrophages (in particular M2 macrophages in 5E) in healthy muscles. Infiltrating cells were not observed in healthy muscles. Is this muscle really healthy? 

RESPONSE: We acknowledge the need for clarification regarding our comparative analysis. Our starting point was the Eisenberg study's findings on miR-155 expression in DMD, noting its significant dysregulation in various myopathies, particularly DMD. Using this data, we aimed to elucidate the specific alterations induced by miR-155 within muscle cells. To achieve this, we conducted in vitro experiments and RNA sequencing to delineate the mRNA expression profile influenced by miR-155. Upon identifying the dysregulated mRNAs, we then compared this profile with that of muscle tissues from DMD patients. This comparison was critical to determine the commonalities between miR-155 targets and the mRNA alterations observed in dystrophic muscles. Regarding macrophages, digital cytometry analysis used transcriptome data from control and muscular dystrophy patients. Based on the mRNA profile, the algorithm results in an enrichment of possible immunological cells and how this enrichment pattern may differ between the groups analyzed. It should be noted that the samples in the control group were obtained from healthy individuals.

  1. In figure 6, they also performed the integrated analysis using protein-protein interaction. It is also difficult to reach a conclusion only by transcriptional data, because miRNAs have been known as key players in gene regulation that target specific mRNAs for degradation or translational repression. To complete integrated analyses, translation efficacy showed be included by analyses of ribosome profiling or protein data.  

RESPONSE:  Thank you for your insightful comment regarding Figure 6. We appreciate your suggestion to enhance the study by including analyses of translational efficacy through methods like ribosome profiling or proteomic data. We agree that miRNAs are crucial in gene regulation, often targeting specific mRNAs for degradation or translational repression, and that transcriptional data alone may not fully capture the nuances of these regulatory mechanisms. In the current study, our focus was primarily on the transcriptional level, considering the resources of our research. However, we acknowledge the potential added value that analyzing translational changes and protein levels could bring to our understanding of miRNA-mediated regulation. While ribosome profiling or comprehensive proteomic analyses were beyond the scope of this study, we believe they represent important avenues for future research. Such analyses would undoubtedly provide a more complete picture of the regulatory effects of miRNAs and contribute to a deeper understanding of their role in the context of muscle cell differentiation and related pathologies. We hope to explore these aspects in future studies to further elucidate the intricate regulatory networks, and we appreciate the reviewer's suggestion in guiding our future research direction.

Reviewer 3 Report

Comments and Suggestions for Authors

The study by Lopes and colleagues represents an interesting exploration of the role of miR-155 based on the experimental and bioinformatics modeling approaches. Overall, the study is quite sound, but the findings beyond transcriptomics are mostly speculative. At least a single experiment to follow up the bioinformatics results would greatly strengthen the study. Apart from this, I have a few minor comments:

1. Please check the ST2 for the standard use of English.

2. Line 118: the phrase "in silico" should be in a different position.

3. For the overlaps of gene sets, please indicate the P-values and the expected number of overlapping genes.

4. Line 197-198: are "shared targets" actually DEGs? Please distinguish between the miRNA targets and regulated genes.

5. Lines 205-206:  Genes belonging to particular biological processes are not "targets".

6. Lines 240-241: it is unclear what protects against senescence.

7. Figure 6B is instructive in principle but can be somewhat misleading since the miRNA operates at the RNA level.

Comments on the Quality of English Language

Some of the wording/sentence structure is problematic or some of the words do not exist in English. The major issues are indicated in the comments to authors. 

Author Response

Reviewer 3

The study by Lopes and colleagues represents an interesting exploration of the role of miR-155 based on the experimental and bioinformatics modeling approaches. Overall, the study is quite sound, but the findings beyond transcriptomics are mostly speculative. At least a single experiment to follow up the bioinformatics results would greatly strengthen the study. Apart from this, I have a few minor comments:

  1. Please check the ST2 for the standard use of English.

RESPONSE: We thank the reviewer for raising this point. We have already checked the English of Supplementary Tables.

  1. Line 118: the phrase "in silico" should be in a different position.

RESPONSE: Thank you, we corrected this issue.

  1. For the overlaps of gene sets, please indicate the P-values and the expected number of overlapping genes.

RESPONSE: We thank the reviewer for this 

valuable suggestion. Following this, we conducted the suggested analysis and have now incorporated the p-values into the manuscript. These can be located on page 3, lines 117 and 119, detailing the overlapping genes in myoblasts and myotubes. Additionally, for the DMD and miR-155 experiments, details regarding the expected number of overlapping genes and the corresponding p-values are presented on page 7, line 192.

  1. Line 197-198: are "shared targets" actually DEGs? Please distinguish between the miRNA targets and regulated genes. AND 5. Lines 205-206:  Genes belonging to particular biological processes are not "targets".

RESPONSE: We appreciate the reviewer's attention to these details and have made the necessary corrections accordingly.

  1. Lines 240-241: it is unclear what protects against senescence.

RESPONSE: We thank the reviewer. We rewrote the phrase to clarify this information (line 235).

  1. Figure 6B is instructive in principle but can be somewhat misleading since the miRNA operates at the RNA level.

RESPONSE: We agree with the reviewer's observation that the current figure could potentially lead to misconceptions about miRNA regulation at the RNA level. This scheme aimed to illustrate the cellular sites of molecules implicated in the regulation by miR-155, specifically regarding the ultimate outcome of this regulatory process. We agree with the reviewer that the scheme may generate a misinterpretation of the regulation of miRNAs at the RNA level. The purpose of this figure was to demonstrate the cellular location of the molecules involved in regulation by miR-155. We felt that schematizing in this way would be more didactic for the bedside, facilitating the visualization of intracellular locations that could be affected by the regulation of miR-155. Unfortunately, the representation of this information is limited to this format at the functional level of the protein.

Some of the wording/sentence structure is problematic or some of the words do not exist in English. The major issues are indicated in the comments to authors. 

RESPONSE: We sincerely appreciate the reviewer's insightful suggestions. We believe we have addressed the key points raised. The review process has been immensely valuable and has notably enhanced the quality of our article.

Round 2

Reviewer 1 Report

Comments and Suggestions for Authors

Comments made during the review were adequately addressed in the revised manuscript. Only some minor mistakes remain and need to be corrected:

line 30: muscle cells

line 82: of miR-155 in

line 84: typing of LGMD must be adjusted to new nomenclature

line 111: myoblasts and myotubes

line 120: repetitive sentences, please rephrase

line 130: involved in

supplement page 3: experimental design

Comments on the Quality of English Language

See language mistakes in the comments. Please check the manuscript throughout.

Author Response

Reviewer 1

Comments made during the review were adequately addressed in the revised manuscript. Only some minor mistakes remain and need to be corrected:

RESPONSE: We are grateful for your careful assessment and appreciate your positive evaluation of our revisions. We have carefully reviewed your comments and have made the necessary corrections to address the identified minor issues.

line 30: muscle cells

RESPONSE: Original: "Through in silico and in vitro approaches, we identify a distinct transcriptional profile induced by miR-155-5p in muscle cell."

Corrected: " Through in silico and in vitro approaches, we identify a distinct transcriptional profile induced by miR-155-5p in muscle cells."

Rationale: We have corrected the sentence to use the plural form of "muscle cells" to better reflect the scientific usage of the term.

line 82: of miR-155 in

RESPONSE: Original: In this study, the authors noted a differential expression in miR-155 in muscle samples from patients (...)

Corrected: In this study, the authors noted a differential expression of miR-155 in muscle samples from patients (...)

Rationale: We have replaced the "in" with the possessive pronoun "of" to more accurately convey the ownership of the expression.

line 84: typing of LGMD must be adjusted to new nomenclature

RESPONSE: Original: (...) patients with Duchenne muscular dystrophy, facioscapulohumeral muscular dystrophy, limb-girdle muscular dystrophy types 2A and 2B, Miyoshi myopathy, nemaline myopathy, polymyositis, dermatomyositis, and inclusion body myositis.

Corrected: (...) patients with Duchenne muscular dystrophy, facioscapulohumeral muscular dystrophy, limb-girdle muscular dystrophies R1 and R2, Miyoshi myopathy, nemaline myopathy, polymyositis, dermatomyositis, and inclusion body myositis.

Rationale: We have updated the typing of LGMD to reflect the current nomenclature, which utilizes the R1 and R2 designations. This change has been made in Table 1 and line 360 as well.

line 111: myoblasts and myotubes

RESPONSE: Original: (...) we next sought to evaluate the profile of gene expression in myoblast and myotube treated with mimic-miR-155.

Corrected: (...) we next sought to evaluate the profile of gene expression in myoblasts and myotubes treated with mimic-miR-155.

Rationale: We have corrected the grammatical error by making "myoblast and myotube" plural.

line 120: repetitive sentences, please rephrase

RESPONSE: Original: We predicted the direct miR-155 targets using the miRWalk, miRTarBase, and TargetScan algorithms. We identified 511 transcripts predicted to be direct targets of miR-155.

Corrected: Using the miRWalk, miRTarBase, and TargetScan algorithms, we identified 511 transcripts predicted to be direct targets of miR-155.

Rationale: We have rephrased the sentence to eliminate the repetition of "predicted" and to improve the overall flow.

line 130: involved in

RESPONSE: Original: The identified kinases involved with the cell cycle progression, inflammation, and fibrogenesis are shown in Supplementary Figure 2.

Corrected: The identified kinases involved in the cell cycle progression, inflammation, and fibrogenesis are shown in Supplementary Figure 2.

Rationale:  The change from "involved with" to "involved in" is indeed more accurate and precise, as it emphasizes the direct role of the kinases in the cellular processes mentioned.  

supplement page 3: experimental design

RESPONSE: Original: experimental desing

Corrected: experimental design

Rationale: We have corrected the spelling error to ensure the accuracy of the term "experimental design."

Reviewer 2 Report

Comments and Suggestions for Authors

The authors answered my concerns well and the revised manuscript is improved much. I do not have any more commnents on it.

Author Response

Reviewer 2

The authors answered my concerns well and the revised manuscript is improved much. I do not have any more comments on it.

RESPONSE: We are grateful for your careful consideration of our manuscript and appreciate your positive assessment of our revisions. We are confident that the manuscript has been substantially improved in response to your comments.
